# Identification of Novel Candidate CD8^+^ T Cell Epitopes of the SARS-CoV2 with Homology to Other Seasonal Coronaviruses

**DOI:** 10.3390/v13060972

**Published:** 2021-05-24

**Authors:** Pradeep Darshana Pushpakumara, Deshan Madhusanka, Saubhagya Dhanasekara, Chandima Jeewandara, Graham S. Ogg, Gathsaurie Neelika Malavige

**Affiliations:** 1Centre for Dengue Research, Department of Immunology and Molecular Medicine, University of Sri Jayewardenepura, Nugegoda 10250, Sri Lanka; mail2medarshana@gmail.com (P.D.P.); deshan993@gmail.com (D.M.); saubhagyadanasekara@gmail.com (S.D.); jeewandara@sjp.ac.lk (C.J.); 2MRC Human Immunology Unit, MRC Weatherall Institute of Molecular Medicine, Radcliffe Department of Medicine, University of Oxford, Oxford OX1 4BH, UK; graham.ogg@ndm.ox.ac.uk

**Keywords:** CD8^+^ T cell epitopes, SARS-CoV2, seasonal coronaviruses, HLA alleles, OC-43, HKU-1, NL-63

## Abstract

Cross-reactive T cell immunity to seasonal coronaviruses (HCoVs) may lead to immunopathology or protection during SARS-CoV2 infection. To understand the influence of cross-reactive T cell responses, we used IEDB (Immune epitope database) and NetMHCpan (ver. 4.1) to identify candidate CD8^+^ T cell epitopes, restricted through HLA-A and B alleles. Conservation analysis was carried out for these epitopes with HCoVs, OC43, HKU1, and NL63. 12/18 the candidate CD8^+^ T cell epitopes (binding score of ≥0.90), which had a high degree of homology (>75%) with the other three HCoVs were within the NSP12 and NSP13 proteins. They were predicted to be restricted through HLA-A*2402, HLA-A*201, HLA-A*206, and HLA-B alleles B*3501. Thirty-one candidate CD8^+^ T cell epitopes that were specific to SARS-CoV2 virus (<25% homology with other HCoVs) were predominantly identified within the structural proteins (spike, envelop, membrane, and nucleocapsid) and the NSP1, NSP2, and NSP3. They were predominantly restricted through HLA-B*3501 (6/31), HLA-B*4001 (6/31), HLA-B*4403 (7/31), and HLA-A*2402 (8/31). It would be crucial to understand T cell responses that associate with protection, and the differences in the functionality and phenotype of epitope specific T cell responses, presented through different HLA alleles common in different geographical groups, to understand disease pathogenesis.

## 1. Introduction

Infection due to the SARS-CoV2 virus is currently the leading cause of mortality among elderly and vulnerable groups in many countries. Outbreaks of COVID-19 are now reported in 220 countries in the world [1], with some countries experiencing a massive second wave. The majority of those who are infected with SARS-CoV2 experience asymptomatic or mild illness, whereas severe illness and fatalities are seen in older individuals and in those with comorbidities such as diabetes, cancer, and cardiovascular disease [2,3,4]. However, the observed case fatality ratios (CFRs) vary widely between countries. For instance, countries such as Mexico, Italy, and Iran report CFRs of over 3.5%, while countries such as India, Turkey, South Korea report CFRs of <2% [5,6]. The deaths per 100,000 population also varies widely among countries, with many European countries and the US reporting rates of over 100 deaths/100,000 population, while the majority of South Asian and South East Asian countries report death rates <25 deaths/100,000 population, despite the ongoing large outbreaks in the South East Asian region [5].

There have been many factors that have been shown to influence the CFRs in different countries including the proportion of the population aged >70 years of age, the GDP per capita, BCG vaccination status [7,8], climate, population density, and social distancing measures, which could account for the differences in the CFRs. However, there could be many immune factors that also lead to these differences, including immunity to human seasonal coronaviruses (HCoVs), providing cross protection against SARS-CoV2. Children are more frequently exposed to HCoVs compared to adults and recently it was shown that a large proportion of children, who were sampled between the years of 2011 to 2018, between the ages of 1 to 16, had IgG antibodies that cross-react with the spike protein of SARS-CoV-2 [9]. Sera from SARS-CoV2 uninfected individuals who had such cross-reactive antibodies were able to neutralize the pseudotypes of SARS-CoV-2 without causing antibody dependent enhancement of infection [9].

Higher lymphocyte counts and specifically CD8^+^ T cell counts have been associated with early viral clearance and reduced disease severity [10]. Robust SARS-CoV2 specific CD4^+^ and CD8^+^ T cell responses have been detected following natural infection [11,12]. SARS-CoV2 cross reactive T cells have been detected following SARS-CoV1 infection in 2003, which were predominantly directed against the N and structural proteins (NSP7 and NSP13) [13]. Interestingly, SARS-CoV2 cross reactive CD4^+^ T cells have also been detected in a large proportion of unexposed individuals in some studies, which are thought to be due the presence of T cells specific to HCoVs [12,14]. However, differences in T cell methodology have resulted in differential findings of cross-reactivity related to altered sensitivity and specificity. Ultimately, it will be important to know if T cells can respond to virally infected target cells where naturally processed epitopes are presented. Nevertheless, the presence of such cross-reactive T cell responses has been speculated to protect against infection with the SARS-CoV2 virus and have also speculated to cause disease pathogenesis or response to vaccines [13,14].

As there is a difference in the CFRs among different countries, which are due to multiple factors, it would be important to understand if prior immunity to HCoVs could influence disease outcome when infected with the SARS-CoV2 virus. The predominant CD8^+^ and CD4^+^ T cell epitopes recognized by a population are also influenced by the frequency of different HLA types of a population. Therefore, in this study, we analyzed the cross reactivity of all the proteins of the SARS-CoV2 virus with three HCoVs (OC43, NL63, and HKU1) and those that are specific to the SARS-CoV2 virus and then used immune epitope database (IEDB) to predict the CD8^+^ T cell epitopes predicted to be restricted through the common HLA-Class I alleles in the Sri Lankan population.

## 2. Materials and Methods

### 2.1. Data Collection for Seasonal Human Corona Virus (HCoV) Sequence Analysis

The full-length protein sequences of the SARS-CoV2 (NC_045512), OC43 (NC_006213), HKU1 (NC_006577), and NL63 (NC_005831) were obtained from the National Center for Biotechnology Information (NCBI) GenBank (https://www.ncbi.nlm.nih.gov/, 5 October 2020). Four structural and 16 non-structural proteins (NSPs) were individually used for T cell epitope prediction and conservation analysis.

### 2.2. T-Cell Epitope Prediction for MHC Class I HLA Alleles

IEDB (Immune epitope database) (www.iedb.org, 10 October 2020) [15] along with NetMHCpan (ver. 4.1) [16] was used for finding putative peptide sequences that were restricted through the MHC Class-I HLA alleles, as these methods use algorithms (generated by artificial neural networks), which have accuracy of epitope prediction. HLA-A*02, HLA-A*24, HLA-A*33, HLA-B*35, HLA-B*40, HLA-B*44, HLA-C*07, HLA-C*06, HLA-C*04, and HLA-C*03 alleles were used for this epitope prediction, as these five alleles are the predominant alleles (each allele seen in >10% in the population) in the Sri Lankan population [17]. The epitope predictions were carried out for 8mer, 9mer, and 10mer epitopes. According to the epitope prediction NetMHCpan EL 4, which gives a score ranging from 0.0 to 1.0. Epitopes with a higher score are considered as stronger binders.

### 2.3. Conservation Analysis of SARS-CoV-2 Proteins and Predicted Epitopes with OC43, HKU1, and NL63

The epitopes which had a predicted binding score of >0.8 through the above approach and were highly conserved between the SARS-CoV2 and OC43, HKU1, and NL63, were further analyzed to determine percentage of identities and similarities between SARS-CoV2 and other HCoVs. The sequences were aligned using the Clustal W on MEGA X software (www.megasoftware.net, 5 October 2020). Conservation of protein sequences were analyzed by using conservation analysis tool available at European Bioinformatics Institute (EBI) (www.ebi.sc.uk, 12 October 2020), IEDB, and Jalview (www.jalview.org, 15 October 2020).

## 3. Results

In order to identify the possible cross-reactive regions between SARS-CoV2 and other HCoVs, we carried out conservation analysis of SARS-CoV-2 virus with OC43, HKU1, and NL63, which have been seen in Sri Lanka. We analyzed the homology and conservation of the 4 structural and the 16 NSPs of SARS-CoV2 with the 3 other HCoVs (Appendix A). The NSP12, NSP13, and NSP16 of the SARS-CoV2 showed ≥65% homology with OC43 and HKU, which are beta coronaviruses, and a >55% homology with NL63, which is an alpha coronavirus.

In contrast, the NSP1 and NSP2 proteins of the SARS-CoV2 showed <20% homology with other three viruses, suggesting that these proteins were more specific for SARS-CoV-2 compared to the other proteins. All four structural proteins (S, E, M, and N), NSP3, and NSP6 showed <35% homology with OC43, HKU1, and NL63. Therefore, immune responses directed at these proteins may also be specific to the SARS-CoV2, unlike immune responses generated against NSP12, NSP13 and NSP16. The SARS-CoV-2 proteins S, E, M, N, and the non-structural proteins showed less homology with NL63 than OC43, and HKU1, suggesting that the. SARS-CoV-2 virus is genetically closer to OC43 and HKU1 than NL63.

### 
3.1. Identification of Possible CD8^+^ Epitopes of the SARS-CoV2 Virus Restricted through HLA-A Alleles of the SARS-CoV2


The NetMHCpan EL4 epitope prediction tool gives peptide binding scores ranging from 0 to 1.0 and we considered a predicted score ≥0.90 for a peptide as indicative of a stronger binder. None of the 8mer peptides gave a predictive score of ≥0.90. However, 39 9mer peptides gave a high binding score of ≥0.90, which were restricted through different HLA-A alleles, (Table 1). Five of these epitopes were identified from the spike and NSP3 proteins and 4 epitopes each were identified from NSP4, NSP6, NSP12, NSP14, and NSP15. Peptides with high binding scores were not identified from the envelope, NSP1, NSP9, NSP10, NSP11, and NSP16 proteins. The epitopes from NSP3 ^726^YYTSNPTTF^734^ (predicted to be restricted through HLA-A*2402) and NSP6 ^70^FLLPSLATV^78^ (predicted to be restricted through HLA-A*0201) gave a score of 0.99, while ^1349^NYMPYFFTL^1357^ from NSP3, ^420^FLLNKEMYL^428^ from NSP4, and ^152^ALWEIQQVV^160^ from NSP8 also gave a score of 0.98 scores. However, they had <45% homology with the other three viruses. 23/39 of the 9mer epitopes predicted in this study were restricted through HLA-A*02 while 16/39 9mer peptides were predicted to be restricted through HLA-A*24. Although HLA-A*33 was an allele seen in over 10% of the Sri Lankan population, 9mers that had a score of ≥0.90 were not identified.

Only four 10mer peptides were predicted to have a score of ≥0.90, two were from spike, and one each from NSP3 and NSP6 proteins (Table 1).

### 
3.2. Identification of Possible CD8^+^ Epitopes of the SARS-CoV2 Virus Restricted through HLA-B Alleles of the SARS-CoV2


As for HLA-A alleles, we considered a predicted score ≥0.90 for a peptide as indicative of a stronger binder. None of the 8mer peptides were found to give a score of ≥0.90 and therefore, were not predicted to be restricted through HLA-B alleles. However, 38 9mer peptides were identified, which had high binding scores and were predicted to be restricted through HLA-B alleles (Table 2). The highest number of epitopes were predicted from spike protein (5/38) and NSP13 (4/38). Three epitopes were predicted from each of the following proteins: namely the nucleocapsid, NSP2, NSP3, NSP4, and NSP12. No epitopes were identified from envelope, membrane, and NSP11 proteins. Nine epitopes gave a score of 0.99 and were ^895^IPFAMQMAY^903^ from spike, ^325^TPSGTWLTY^333^ from nucleocapsid, ^195^SEVGPEHSL^203^ and ^562^GETLPTEVL^570^ both from NSP2, ^120^EEFEPSTQY^128^ and ^546^QEILGTVSW^554^ both from NSP3, ^72^LPSLATVAY^80^ from NSP6, ^4^SEFSSLPSY^12^ from NSP8, and ^608^VENPHLMGW^616^ from NSP12.

17/38 of these epitopes were predicted to be restricted through HLA-B*35 11/38 through HLA-B*40 and 10/38 through HLA-B*44. While most of these peptides showed <45% homology with OC43, NL63 and HKU-1, ^141^TEETFKLSY^149^ and ^608^VENPHLMGW^616^ restricted through HLA-B*44, ^337^VPFVVSTGY^345^ restricted through HLA-B*35 and ^161^RELHLSWEV^169^ restricted through HLA-B*40 showed a homology of >75% with OC43 and HKU-1, which are two other beta coronaviruses.

Only 18 10mers peptides were predicted to have a score of ≥0.90 that were restricted through HLA-B (Table 2), 4/18 were identified NSP3. 10mer peptides with high binding scores were not predicted from envelope, nucleocapsid, NSP1, NSP4, NSP5, NSP6, NSP9, NSP11, and NSP16 9 proteins. 3/18 of these predicted 10mers were shown to be restricted through HLA-B*35, 4/18 through HLA-B*40, and 11/18 through HLA-B*44. Therefore, there may be a higher probability for 10mers to bind to HLA-B*44 alleles.

### 
3.3. Identification of Possible CD8^+^ Epitopes of the SARS-CoV2 Virus Restricted through HLA-C Alleles of the SARS-CoV2


None of the 8mer and 10mer peptides were found to give a score of ≥0.90 and therefore, were not predicted to be restricted through HLA-C alleles. However, 21 9mer peptides that were identified had high binding scores and were predicted to be restricted through HLA-C alleles (Table 3). The highest number of epitopes were predicted from NSP3 protein (6/21) and spike (4/21). Epitopes were not predicted from each of the following proteins: namely the nucleocapsid, envelop, NSP2, NSP4, NSP6, NSP9, NSP11, NSP12, NSP15 and NSP16.

### 3.4. Identification of CD8^+^ T Cell Epitopes of SARS-CoV2, Which Show ≥75% Homology with OC43, HKU1, and NL63

After identification of peptides that had high predicted values to be restricted through the common HLA-A, B, and C alleles present in the Sri Lankan population, we proceeded to identify the regions of the SARS-CoV2 virus, which had a >75% homology with the HCoVs. We then proceeded to identify CD8^+^ T cell epitopes within these regions, which were candidates to be restricted through these HLA alleles. This was to determine if we could identify CD8^+^ T cell epitopes of the SARS-CoV2, which were likely to cross-react with the other HCoVs. None of the predicted CD8^+^ 8mer epitopes identified within the SARS-CoV2 virus gave a high binding score and therefore, only predicted 9mer and 10mer CD8^+^ T cell epitopes of the SARS-CoV2 virus were analyzed for the degree of homology with OC43, HKU1, and NL63.

Epitopes that were identified to have a ≥75% homology with more than two HCoV viruses are shown in Table 4. Thirty-four 9mer epitopes and 18 10mer peptides identified within the SARS-CoV2 virus had ≥75% homology with ≥2 HCoV viruses. Of the 9mer peptides, 22/34 epitopes gave a peptide binding score of ≥0.90. 11/34 of these CD8^+^ T cell epitopes within these cross-reactive regions were predicted to be restricted through HLA-A, 6/34 were predicted to be restricted through HLA-B alleles and 5/34 were predicted to be restricted through HLA-C alleles. Six highly cross-reactive CD8^+^ T cell epitopes (9mers) with high HLA-A (A*201 and A*206) binding scores were identified from the NSP12 and NSP13 (^334^FVDGVPFVV^342^). Two of these peptides showed 100% homology with OC43 and HKU1. The Envelope, nucleocapsid proteins and the other non-structural proteins (apart from NSP12 and NSP13) did not have regions with >75% homology with the other HCoVs. The alignment of SARS-CoV2 NSP12 and NSP13, in which most of the cross-reactive epitopes were identified from and their position are shown in Appendix A.

Of the 10mer peptides analyzed, 18 were identified within SARS-CoV2 to have ≥75% homology with ≥2 HCoVs (Table 4). Only one 10mer peptide identified within NSP-13 (^446^AEIVDTVSAL^455^) gave a score of ≥0.90 score. 14/18 of these 10mer CD8^+^ T cell epitopes were predicted to be restricted through HLA-A and 3/18 were predicted to be restricted through HLA-B alleles. Only one epitope out of 18 was predicted to be restricted through HLA-C. As with the 9mer peptides, 6 of the 10mer peptides, which were highly homologous with OC43, HKU1 and NL63 were found within the NSP12 and NSP13 region.

### 3.5. Identification of CD8^+^ T cell Epitopes of SARS-CoV2, Which Show ≤25% Homology with OC43, HKU1, and NL63

After identification of highly cross reactive CD8^+^ T cell epitopes within the SARS-CoV2, we proceeded to identify regions, which were specific to the virus and did not cross react with other HCoV2, and therefore, are likely to be SARS-CoV2 specific CD8^+^ T cell epitopes. 9mer peptides of the representing different regions of the SARS-CoV2 virus, which have ≤25% homology with >2 HCoV viruses were analyzed and 60 such potential CD8^+^ T cell epitopes were identified. (Table 5). 19/60 CD8^+^ T cell epitopes were predicted to be restricted through HLA-A alleles, 20/60 epitopes were predicted to be restricted through HLA-B alleles and 21/62 epitopes were predicted to be restricted through HLA-C alleles. 31/60 9mer peptides gave a binding score of ≥0.90. 12/31 of these CD8^+^ T cell epitopes were predicted to be restricted through HLA-A alleles, 14/31 predicted to be restricted through HLA- B alleles and 5/31 predicted to be restricted through HLA- C alleles. A region within the spike protein (^686^VASQSIIAY^694^) had no homology with the other HCoVs but had a high binding score of >0.95 to HLA-B*3501 and two other 9mer peptides within the nucleocapsid (^325^TPSGTWLTY^333^ and ^322^MEVTPSGTW^330^) had <22% homology and were predicted to be restricted through HLA-B*3501 and HLA-B*4403. Three other CD8^+^ T cell epitopes within NSP2, NSP3 and NSP6, which had high binding scores but had 0% homology were also identified.

We identified 49 10mers as CD8^+^ T cell epitopes, which had ≤25% homology with two HCoV viruses (Table 5). 5/44 epitopes gave a score of ≥0.90. 22/49 of these CD8^+^ T cell epitopes were predicted to be restricted through HLA-A, 22/49 were predicated to be restricted through HLA-B alleles and 5/49 were predicated to be restricted through HLA-C alleles. Again, the peptides that had the highest binding scores and least percentage identified were predicted to be restricted through HLA-B*3501 and HLA-B*4403. The highest binders, which were specific to SARS-CoV2, were identified within the spike protein (^95^TEKSNIIRGW^104^), NSP2 (^489^KEIKESVQTF^498^), and NSP3 (^120^EEEFEPSTQY^129^ and ^502^VPTDNYITTY^511^).

### 3.6. Conservational Analysis of the Candidate CD8^+^ T Cell Epitopes with Binding Scores of ≥0.90

We proceeded to investigate if the 18 candidate CD8^+^ T cell epitopes that had a percentage identity of >75% with other HCoVs and the 31 SARS-CoV2 specific (<25% percentage identity) were conserved within the SARS-CoV2. We found that these candidate epitopes were highly conserved (Appendix A) and these regions were highly conserved within the new SARS-CoV2 variants as well (Appendix A).

### 3.7. Similarity of Candidate Peptides with Published CD8^+^ T Cell SARS-CoV2 Epitopes

Several CD8^+^ T cell epitopes that are restricted through different HLA-A and B alleles have been published [11,18,19,20,21]. We proceeded to find out if any of the candidate CD8^+^ T cell epitopes were already identified in patients who were naturally infected with the SARS-CoV2 virus. We found that 20/31 candidate highly conserved T cell epitopes which were found to be specific to the SARS-CoV2 (<25% homology with other HCoVs) had been identified in infected individuals (Appendix A). In our HLA allele prediction analysis using the dominant HLA alleles in Sri Lanka, although some of the epitopes were predicted to be restricted though HLA-B*3501 and HLA-B*4403, some of these epitopes were found to be restricted through HLA-A*0201, A*1101 and HLA-A*0301.

7/18 of the candidate T cell epitopes, which were found to be cross reactive (>75% homology with the other HCoV2s) were also identified from those who were naturally infected. For the candidate CD8^+^ T cell epitopes that were found to be cross reactive with other HCoV2, the predicted HLA allele by us and the HLA allele restriction identified following natural infection were similar in 4/7 epitopes (Appendix A).

## 4. Discussion

In this study we have identified candidate CD8^+^ T cell epitopes, which were highly conserved within SARS-CoV2, and some which show >75% percentage homology with the HCoV2s OC43, HKU1 and NL63, and therefore, are candidates to give rise to cross-reactive T cell responses. The majority of the predicted CD8^+^ T cell epitopes (binding score of ≥0.90), which had a high degree of homology with the other three HCoV2s were within the NSP12 and NSP13 proteins. They were predicted to be restricted through HLA-A*2402, HLA-A*0201, HLA-A*0206 and HLA-B alleles B*3501. Therefore, the presence of SARS-CoV2 cross reactive CD8^+^ T cell responses could depend on the frequency of the above HLA-A alleles in a population, as the most cross-reactive candidate CD8^+^ T cell epitopes are restricted through these alleles.

The frequency of HLA-A*0201 and HLA-A*0206 in the Sri Lankan population are 4.9% to 6.6% and 2.1% to 2.4% respectively, while the frequency of HLA-A*2402 is 20.8% to 30.3% [17,22]. HLA-B*35 frequency in the Sri Lankan population is 21% to 23% [17,22]. In contrast, the most frequent HLA-A alleles in the European, US and Brazilian populations are HLA-A*0201 and A*0206 (24.5% to 27.5%), which are several fold higher than in the Sri Lankan population, while HLA-A*2402 and B*3501 are lower (7.9% to 9.5%) [23,24,25]. In silico analysis recently showed that HLA-A*0201 was associated with a higher risk of COVID-19, while HLA-A*2402 was shown to associate with higher capacity to present SARS-CoV2 antigens [26]. SARS-CoV2 specific HLA-A*0201 CD8^+^ T cell epitopes were shown to have suboptimal antiviral response and of a reduced frequency when compared to other viral infections such as influenza and Epstein-Barr viral infection [27]. The in-silico analysis showed that countries (Italy, France, Germany, Brazil) in which the most frequent HLA-A allele was A*0201 had the highest COVID-19 case fatality rates (CFRs), whereas those where HLA-A*2402 allele was the most frequent (India, Iran) had lower CFRs [26]. Therefore, it would be important to investigate if certain HLA alleles, presented CD8^+^ T cell epitopes that associate with protection, whereas if certain other alleles present epitopes that are associated with immunopathology and poor antiviral capacity.

In contrast, the candidate CD8^+^ T cell epitopes, which were highly conserved identified within SARS-CoV2 virus that are likely to be specific, were predominantly identified within the structural proteins (spike, envelope, membrane, and nucleocapsid) and the NSP1, NSP2, and NSP3. 6/31 of these candidate CD8^+^ T cell epitopes (binding score of ≥0.90), that were specific to SARS-CoV2 (<25% homology with the HCoVs) were predicted to be restricted through HLA-B*3501, 6/31 through HLA-B*4001, 7/31 through HLA-B*4403 and 8/31 through HLA-A*2402. Only 3/31of the SARS-CoV2 specific candidate T cell epitopes were predicted to be restricted through HLA-A*0201 or A*0206 alleles, common in Europe, USA, and Brazil. Therefore, HLA-A*2402 and HLA-B*3501, HLA-B*4001, and HLA-B*4403, which are predominant HLA alleles in Sri Lanka and India, may present both highly cross-reactive and SARS-CoV2 specific CD8^+^ T cell epitopes. Indeed, our analysis showed that 20/31 highly conserved, SARS-CoV2 specific candidate CD8^+^ T cell epitopes were already identified in those with natural infection. Although our HLA allele prediction using the dominant HLA alleles in Sri Lanka predicted these epitopes to be restricted through HLA-B*3501 and HLA-B*4403, some were found to be presented through different HLA alleles in those who were naturally infected in Europe and USA [11,18,21]. However, given that these epitopes are highly conserved within the virus, it is possible that they could be presented by multiple HLA alleles, which should be further investigated.

T cell responses of higher magnitude and breath have been observed in patients who had more severe COVID-19 [12,28]. However, it is not yet known if a higher magnitude and breath of T cell responses in COVID-19 is associated with protection or immunopathology. There is a debate if cross-reactive T cells cause immunopathology in certain viral infections such as in dengue [29,30] and if such cross-reactive T cells in SARS-CoV2 are protective should be investigated. It is hoped that the candidate epitopes presented here, will be of help in subsequent functional T cell analyses, particularly as viral variants emerge as they were found to be highly conserved within the new UK SARS-CoV2 variant, B.1.1.7 and the new South African variant B.1.351. By focusing on highly conserved regions within and between each coronavirus, the candidate epitopes may be of value in understanding immune responses across populations and for future vaccine design. In addition, since many vaccines for COVID-19 are currently been rolled out, it would be crucial to understand T cell responses that associate with protection and the differences in the functionality of epitope specific T cell responses, presented through different HLA alleles.

## 5. Conclusions

In summary, we have identified candidate SARS-CoV2 CD8^+^ T cell responses that are highly cross reactive with other HCoVs and that also are specific to SARS-CoV2. Cross-reactive epitopes were predominantly identified from NSP12 and NSP13, while specific epitopes were identified within the structural proteins and NSP1–3. It would be crucial to understand the CD8^+^ T cell epitopes presented through the most frequent HLA alleles, their phenotype and functionality to better understand the immune responses to SARS-CoV2 and possible implications for vaccines.

## Figures and Tables

**Table 1 viruses-13-00972-t001:** Predicted CD8^+^ epitopes of the SARS-CoV2 virus restricted through HLA-A alleles.

9mer Peptides with a Peptide Binding Score of ≥0.90						
Protein	HLA-A Allele	Sequence	Score	OC43 % Identity with SARS-CoV2	HKU1 % Identity with SARS-CoV2	NL63 % Identity with SARS-CoV2
Spike	A*02:01	^269^YLQPRTFLL^277^	0.97	44	44	22
Spike	A*24:02	^1208^QYIKWPWYI^1216^	0.95	77	77	77
Spike	A*02:01	^976^VLNDILSRL^984^	0.94	67	67	33
Spike	A*24:02	^635^VYSTGSNVF^643^	0.93	22	22	22
Spike	A*02:01	^109^TLDSKTQSL^117^	0.91	22	22	33
Membrane	A*24:02	^95^YFIASFRLF^103^	0.91	77	77	77
Nucleocapsid	A*02:01	^222^LLLDRLNQL^230^	0.96	33	33	33
NSP2	A*02:01	^265^GLNDNLLEI^273^	0.92	44	33	44
NSP2	A*24:02	^497^TFFKLVNKF^505^	0.90	33	22	22
NSP3	A*24:02	^726^YYTSNPTTF^734^	0.99	22	22	22
NSP3	A*24:02	^1349^NYMPYFFTL^1357^	0.98	33	22	11
NSP3	A*24:02	^816^YYHTTDPSF^824^	0.96	11	11	0
NSP3	A*24:02	^364^LYDKLVSSF^372^	0.95	33	33	11
NSP3	A*24:02	^1081^YYKKDNSYF^1089^	0.94	33	33	0
NSP4	A*02:01	^420^FLLNKEMYL^428^	0.98	33	33	33
NSP4	A*02:01	^359^FLAHIQWMV^367^	0.94	44	44	33
NSP4	A*24:02	^351^FYLTNDVSF^359^	0.92	22	22	22
NSP4	A*24:02	^486^LYQPPQTSI^494^	0.90	66	66	66
NSP5	A*02:06	^194^LIQDYIQSV^202^	0.95	44	33	11
NSP6	A*02:01	^70^FLLPSLATV^78^	0.99	33	33	44
NSP6	A*02:01	^141^TLMNVLTLV^149^	0.92	0	0	0
NSP6	A*24:02	^84^VYMPASWVM^92^	0.91	0	0	11
NSP6	A*24:02	^115^MYASAVVLL^123^	0.90	22	22	22
NSP7	A*02:01	^12^VLLSVLQQL^20^	0.95	66	66	44
NSP8	A*02:01	^152^ALWEIQQVV^160^	0.98	44	33	22
NSP12	A*24:02	^37^IYNDKVAGF^45^	0.95	66	44	44
NSP12	A*02:01	^123^TMADLVYAL^131^	0.93	77	77	77
NSP12	A*02:06	^334^FVDGVPFVV^342^	0.93	100	100	88
NSP12	A*02:01	^854^LMIERFVSL^862^	0.91	88	88	55
NSP13	A*02:01	^ 239 ^ TLVPQEHYV^247^	0.95	77	77	44
NSP13	A*24:02	^ 397 ^ VYIGDPAQL^405^	0.91	100	100	77
NSP14	A*02:01	^321^LLADKFPVL^329^	0.94	11	11	33
NSP14	A*02:01	^176^NLSDRVVFV^184^	0.93	66	66	77
NSP14	A*02:01	^184^VLWAHGFEL^192^	0.92	66	44	66
NSP14	A*02:01	^494^YLDAYNMMI^502^	0.91	44	44	44
NSP15	A*02:06	^243^SQLGGLHLL^251^	0.96	66	66	66
NSP15	A*02:01	^297^LLLDDFVEI^305^	0.95	66	88	88
NSP15	A*02:06	^312^SVVSKVVKV^320^	0.94	66	66	55
NSP15	A*02:06	^181^KVDGVVQQL^189^	0.92	22	22	11
**10mer Peptides with a Peptide Biding Score of ≥0.90**						
**Protein**	**HLA-A Allele**	**Sequence**	**Sore**	**OC43 % Identity with SARS-CoV2**	**HKU1 % Identity with SARS-CoV2**	**NL63 % Identity with SARS-CoV2**
Spike	A*24:02	^1066^TYVPAQEKNF^1075^	0.94	30	30	10
Spike	A*24:02	^159^VYSSANNCTF^168^	0.90	10	30	20
NSP3	A*24:02	^717^VYYTSNPTTF^726^	0.98	30	40	20
NSP6	A*24:02	^242^YDYLVSTQEF^251^	0.91	50	50	50

**Table 2 viruses-13-00972-t002:** Predicted CD8^+^ epitopes of the SARS-CoV2 virus restricted through HLA-B alleles.

9mer Peptides with a Peptide Binding Score of ≥0.90						
Protein	HLA-B Allele	Sequence	Score	OC43 % Identity with SARS-CoV2	HKU1 % Identity with SARS-CoV2	NL63 % Identity with SARS-CoV2
Spike	B*35:01	^895^IPFAMQMAY^903^	0.99	33	44	33
Spike	B*35:01	^83^LPFNDGVYF^91^	0.98	22	33	11
Spike	B*44:03	^1200^QELGKYEQY^1208^	0.98	44	44	33
Spike	B*35:01	^686^VASQSIIAY^694^	0.98	0	0	0
Spike	B*40:01	^1015^AEIRASANL^1023^	0.98	22	22	44
Nucleocapsid	B*35:01	^325^TPSGTWLTY^333^	0.99	22	22	22
Nucleocapsid	B*44:03	^322^MEVTPSGTW^330^	0.96	11	11	0
Nucleocapsid	B*35:01	^79^SPDDQIGYY^87^	0.92	22	22	55
NSP1	B*40:02	^56^VEKGVLPQL^64^	0.98	22	22	11
NSP1	B*35:01	^110^HVGEIPVAY^118^	0.95	22	11	11
NSP2	B*40:01	^195^SEVGPEHSL^203^	0.99	11	11	0
NSP2	B*40:01	^562^GETLPTEVL^570^	0.99	0	0	0
NSP2	B*44:02	^52^REHEHEIAW^60^	0.98	22	22	22
NSP3	B*44:03	^120^EEFEPSTQY^128^	0.99	11	11	0
NSP3	B*44:03	^546^QEILGTVSW^554^	0.99	22	22	0
NSP3	B*40:01	^1799^AELAKNVSL^1807^	0.98	0	0	0
NSP4	B*40:02	^309^GEYSHVVAF^317^	0.98	44	44	33
NSP4	B*35:01	^373^VPFWITIAY^381^	0.98	33	44	00
NSP4	B*35:01	^174^NVLEGSVAY^182^	0.97	11	11	44
NSP5	B*35:01	^93^TANPKTPKY^101^	0.95	66	66	44
NSP6	B*35:01	^72^LPSLATVAY^80^	0.99	44	44	33
NSP7	B*35:01	^41^LAKDTTEAF^49^	0.91	33	33	22
NSP8	B*44:03	^4^SEFSSLPSY^12^	0.99	44	44	66
NSP8	B*40:01	^47^SEFDRDAAM^55^	0.93	44	44	44
NSP9	B*44:03	^67^TELEPPCRF^75^	0.95	66	66	66
NSP10	B*35:01	^19^FAVDAAKAY^27^	0.98	55	55	55
NSP12	B*44:02	^608^VENPHLMGW^616^	0.99	77	77	88
NSP12	B*40:01	^874^QEYADVFHL^882^	0.98	44	44	44
NSP12	B*35:01	^337^VPFVVSTGY^345^	0.97	88	88	55
NSP13	B*35:01	^291^FAIGLALYY^299^	0.96	66	77	66
NSP13	B*44:03	^141^TEETFKLSY^149^	0.96	77	77	55
NSP13	B*40:01	^155^REVLSDREL^163^	0.95	55	55	44
NSP13	B*40:02	^161^RELHLSWEV^169^	0.91	77	77	66
NSP14	B*35:01	^428^TPAFDKSAF^436^	0.96	44	44	77
NSP14	B*35:03	^19^HPTQAPTHL^27^	0.94	55	55	55
NSP15	B*35:03	^49^LPVNVAFEL^57^	0.96	66	66	88
NSP15	B*44:03	^200^QEFKPRSQM^208^	0.92	33	33	55
NSP16	B*44:02	^141^KENDSKEGF^149^	0.94	55	55	66
**10mer Peptides with a Peptide Binding Score of ≥0.90**						
**Protein**	**HLA-B Allele**	**Sequence**	**Sore**	**OC43 % Identity with SARS-CoV2**	**HKU1 % Identity with SARS-CoV2**	**NL63 % Identity with SARS-CoV2**
Spike	B*44:02	^95^TEKSNIIRGW^104^	0.95	10	0	10
Membrane	B*44:03	^11^EELKKLLEQW^20^	0.94	40	20	30
NSP2	B*44:03	^489^KEIKESVQTF^498^	0.95	0	0	10
NSP3	B*44:03	^120^EEEFEPSTQY^129^	0.98	20	10	0
NSP3	B*44:03	^1072^TEIDPKLDNY^1081^	0.96	40	30	10
NSP3	B*35:01	^502^VPTDNYITTY^511^	0.94	0	0	0
NSP3	B*44:03	^94^GEFKLASHMY^103^	0.93	50	50	0
NSP7	B*40:01	^46^TEAFEKMVSL^55^	0.91	50	40	20
NSP8	B*44:03	^3^ASEFSSLPSY^12^	0.94	40	40	60
NSP10	B*40:01	^5^TEVPANSTVL^14^	0.92	50	60	50
NSP12	B*44:03	^875^QEYADVFHLY^884^	0.99	50	50	40
NSP12	B*44:03	^166^VENPDILRVY^175^	0.95	80	70	80
NSP12	B*44:03	^608^DVENPHLMGW^617^	0.90	80	70	80
NSP13	B*40:01	^446^AEIVDTVSAL^455^	0.96	90	80	80
NSP14	B*44:02	^77^EEAIRHVRAW^86^	0.96	70	60	60
NSP14	B*35:01	^42^IPGIPKDMTY^51^	0.92	20	20	30
NSP15	B*35:01	^269^IPMDSTVKNY^278^	0.96	30	30	10
NSP15	B*40:01	^40^VELFENKTTL^49^	0.95	50	30	60

**Table 3 viruses-13-00972-t003:** Predicted CD8^+^ epitopes of the SARS-CoV2 virus restricted through HLA-C alleles.

9mer Peptides with a Peptide Binding Score of ≥0.90						
Protein	HLA-B Allele	Sequence	Score	OC43 % Identity with SARS-CoV2	HKU1 % Identity with SARS-CoV2	NL63 % Identity with SARS-CoV2
Spike	HLA-C*04:01	^1137^VYDPLQPEL^1145^	0.99	22	22	11
Spike	HLA-C*06:02	^327^VRFPNITNL^335^	0.98	11	11	0
Spike	HLA-C*03:02	^687^VASQSIIAY^695^	0.97	11	11	0
Spike	HLA-C*03:02	^1054^QSAPHGVVF^1062^	0.93	55	55	55
Membrane	HLA-C*03:02	^ 37 ^ FAYANRNRF^45^	0.96	44	44	22
Membrane	HLA-C*03:02	^ 170 ^ VATSRTLSY^178^	0.96	33	33	44
NSP1	HLA-C*03:02	^ 77 ^ RTAPHGHVM^85^	0.90	11	11	11
NSP3	HLA-C*03:02	^1776^YVNTFSSTF^1784^	0.96	66	66	33
NSP3	HLA-C*04:01	^734^TFDNLKTLL^742^	0.96	11	11	11
NSP3	HLA-C*03:02	^1651^VARDLSLQF^1659^	0.95	44	33	33
NSP3	HLA-C*04:01	^1772^MFDAYVNTF^1780^	0.95	77	77	33
NSP3	HLA-C*04:01	^364^LYDKLVSSF^372^	0.94	33	33	11
NSP3	HLA-C*03:02	^1735^SAKSASVYY^1743^	0.90	33	33	22
NSP5	HLA-C*03:02	^ 93 ^ TANPKTPKY^101^	0.96	66	66	44
NSP7	HLA-C*03:02	^ 41 ^ LAKDTTEAF^49^	0.92	33	33	22
NSP8	HLA-C*03:02	^ 130 ^ VVIPDYNTY^138^	0.92	44	55	44
NSP10	HLA-C*03:02	^ 19 ^ FAVDAAKAY^27^	0.99	55	55	55
NSP13	HLA-C*03:02	^ 209 ^ VVYRGTTTY^217^	0.95	77	77	44
NSP13	HLA-C*03:02	^ 291 ^ FAIGLALYY^299^	0.94	77	77	55
NSP13	HLA-C*03:02	^ 225 ^ FVLTSHTVM^233^	0.90	77	77	77
NSP14	HLA-C*06:02	^ 162 ^ VRIKIVQML^170^	0.93	77	77	77

**Table 4 viruses-13-00972-t004:** Predicted CD8^+^ epitopes of the SARS-CoV2 virus, which show ≥75% homology with OC43, HKU1, and NL63.

9mer Peptides with ≥75% Homology with OC43, HKU1, and NL63						
Protein	HLA Allele	Sequence	Score	OC43 % Identity with SARS-CoV2	HKU1 % Identity with SARS-CoV2	NL63 % Identity with SARS-CoV2
Spike	A*24:02	^1208^QYIKWPWYI^1216^	0.95	77	77	77
Membrane	A*24:02	^95^YFIASFRLF^103^	0.91	77	77	77
NSP3	C*04:01	^1772^MFDAYVNTF^1780^	0.95	77	77	33
NSP5	A*02:06	^159^FVYMHQLEL^167^	0.76	100	88	66
NSP5	B*35:01	^95^NPKTPKYKF^103^	0.47	77	77	55
NSP10	B*35:03	^36^QPITNCVKM^44^	0.74	88	88	66
NSP12	A*02:01	^123^TMADLVYAL^131^	0.93	77	77	77
NSP12	A*02:06	^334^FVDGVPFVV^342^	0.93	100	100	88
NSP12	A*02:01	^854^LMIERFVSL^862^	0.91	88	88	55
NSP12	A*02:01	^334^FVDGVPFVV^342^	0.90	100	100	77
NSP12	B*44:03	^608^VENPHLMGW^616^	0.99	77	77	88
NSP12	B*35:01	^337^VPFVVSTGY^345^	0.97	88	88	55
NSP12	C*03:02	^534^NVIPTITQM^542^	0.82	77	77	66
NSP12	C*03:02	^340^FVVSTGYHF^348^	0.80	77	77	55
NSP13	A*02:01	^ 239 ^ TLVPQEHYV^247^	0.95	77	77	44
NSP13	A*24:02	^ 397 ^ VYIGDPAQL^405^	0.91	100	100	77
NSP13	A*02:06	^ 239 ^ TLVPQEHYV^247^	0.91	77	77	44
NSP13	B*35:01	^291^FAIGLALYY^299^	0.96	77	77	66
NSP13	B*44:03	^141^TEETFKLSY^149^	0.96	77	77	55
NSP13	B*40:02	^161^RELHLSWEV^169^	0.91	77	77	66
NSP13	C*03:02	^ 209 ^ VVYRGTTTY^217^	0.95	77	77	44
NSP13	C*03:02	^ 291 ^ FAIGLALYY^299^	0.94	77	77	55
NSP13	C*03:02	^ 225 ^ FVLTSHTVM^233^	0.90	77	77	77
NSP13	C*06:02	^ 211 ^ YRGTTTYKL^219^	0.78	88	88	55
NSP14	A*02:01	^176^NLSDRVVFV^184^	0.93	77	66	77
NSP14	B*35:01	^509^WVYKQFDTY^517^	0.65	77	77	55
NSP14	C*06:02	^ 162 ^ VRIKIVQML^170^	0.93	77	77	77
NSP14	C*03:02	^ 487 ^ HANEYRLYL^495^	0.83	77	77	55
NSP15	A*02:01	^297^LLLDDFVEI^305^	0.95	66	88	88
NSP15	B*35:03	^49^LPVNVAFEL^57^	0.96	77	66	88
NSP16	A*02:01	^53^YLNTLTLAV^61^	0.89	88	88	55
NSP16	A*24:02	^46^KYTQLCQYL^54^	0.82	100	100	100
NSP16	A*33:03	^247^MSKFPLKLR^255^	0.78	77	77	44
NSP16	C*04:01	^ 131 ^ MYDPKTKNV^139^	0.83	77	77	44
**10mer Peptides with ≥75% Homology with OC43, HKU1, and NL63**						
**Protein**	**HLA Allele**	**Sequence**	**Sore**	**OC43 % Identity with SARS-CoV2**	**HKU1 % Identity with SARS-CoV2**	**NL63 % Identity with SARS-CoV2**
NSP4	C*03:02	^478^FSNSGSDVLY^487^	0.41	80	70	20
NSP12	A*02:06	^332^KIFVDGVPFV^341^	0.88	90	90	70
NSP12	A*02:01	^332^KIFVDGVPFV^341^	0.88	90	90	70
NSP13	A*24:02	^216^TYKLNVGDYF^225^	0.77	80	70	50
NSP13	A*02:01	^40^KLVLSVNPYV^49^	0.58	80	80	50
NSP13	A*33:03	^381^NYDLSVVNAR^390^	0.54	80	80	80
NSP13	A*33:03	^551^ETAHSCNVNR^560^	0.52	90	90	80
NSP13	B*40:01	^446^AEIVDTVSAL^455^	0.96	90	80	80
NSP13	B*40:02	^446^AEIVDTVSAL^455^	0.87	90	80	80
NSP14	A*33:03	^516^TYNLWNTFTR^525^	0.58	80	80	50
NSP14	A*24:02	^510^VYKQFDTYNL^519^	0.55	80	80	60
NSP15	A*33:03	^52^NVAFELWAKR^61^	0.66	80	60	90
NSP15	A*02:06	^243^SQLGGLHLLI^252^	0.52	70	70	80
NSP16	A*24:02	^241^SYSLFDMSKF^250^	0.75	80	80	50
NSP16	A*33:03	^246^DMSKFPLKLR^255^	0.59	80	70	50
NSP16	A*24:02	^221^GYVMHANYIF^230^	0.55	80	80	70
NSP16	A*02:01	^243^SLFDMSKFPL^252^	0.47	90	80	50

**Table 5 viruses-13-00972-t005:** Predicted CD8^+^ epitopes of the SARS-CoV2 virus, which show ≤25% homology with OC43, HKU1, and NL63.

9mer Peptides with ≤25% Homology with OC43, HKU1, and NL63						
Protein	HLA Allele	Sequence	Score	OC43 % Identity with SARS-CoV2	HKU1 % Identity with SARS-CoV2	NL63 % Identity with SARS-CoV2
Spike	A*24:02	^635^VYSTGSNVF^643^	0.93	22	22	22
Spike	A*02:01	^109^TLDSKTQSL^117^	0.91	22	22	33
Spike	B*35:01	^83^LPFNDGVYF^91^	0.98	22	33	11
Spike	B*35:01	^686^VASQSIIAY^694^	0.98	0	0	0
Spike	B*40:01	^1015^AEIRASANL^1023^	0.98	22	22	44
Spike	C*04:01	^1137^VYDPLQPEL^1145^	0.99	22	22	11
Spike	C*06:02	^327^VRFPNITNL^335^	0.98	11	11	0
Spike	C*03:02	^687^VASQSIIAY^695^	0.97	11	11	0
Spike	C*07:01	^327^VRFPNITNL^335^	0.87	11	11	0
Spike	C*07:02	^327^VRFPNITNL^335^	0.84	11	11	0
Spike	C*06:02	^402^IRGDEVRQI^410^	0.83	11	11	0
Spike	C*04:01	^78^RFDNPVLPF^86^	0.83	11	11	11
Membrane	A*33:03	^138^LVIGAVILR^146^	0.72	22	22	22
Membrane	B*40:01	^136^ SELVIGAVI ^144^	0.73	11	11	11
Envelope	A*33:03	^61^RVKNLNSSR^69^	0.61	11	11	0
Envelope	A*33:03	^30^TLAILTALR^38^	0.60	22	22	11
Envelope	B*40:01	^6^SEETGTLIV^14^	0.55	11	22	11
Envelope	B*35:03	^4^FVSEETGTL^12^	0.43	11	11	11
Nucleocapsid	B*35:01	^325^TPSGTWLTY^333^	0.99	22	22	22
Nucleocapsid	B*44:03	^322^MEVTPSGTW^330^	0.96	11	11	0
Nucleocapsid	C*03:03	^ 403 ^ FSKQLQQSM^411^	0.73	11	11	11
NSP1	A*24:02	^135^SYGADLKSF^143^	0.89	0	0	0
NSP1	A*02:01	^84^VMVELVAEL^92^	0.85	22	22	0
NSP1	B*40:02	^56^VEKGVLPQL^64^	0.98	22	22	11
NSP1	B*35:01	^110^HVGEIPVAY^118^	0.95	22	11	11
NSP1	B*35:01	^89^VAELEGIQY^97^	0.63	11	0	11
NSP1	C*03:02	^ 77 ^ RTAPHGHVM^85^	0.90	11	11	11
NSP1	C*03:02	^ 110 ^ HVGEIPVAY^118^	0.81	22	11	11
NSP1	C*03:02	^ 165 ^ HSSGVTREL^173^	0.71	11	11	0
NSP2	A*24:02	^497^TFFKLVNKF^505^	0.90	33	22	22
NSP2	A*02:06	^420^YITGGVVQL^428^	0.86	22	22	22
NSP2	A*02:06	^439^TVYEKLKPV^447^	0.83	11	11	11
NSP2	B*40:01	^195^SEVGPEHSL^203^	0.99	11	11	0
NSP2	B*40:01	^562^GETLPTEVL^570^	0.99	0	0	0
NSP2	B*44:03	^52^REHEHEIAW^60^	0.98	22	22	22
NSP2	C*06:02	^ 363 ^ VRSIFSRTL^371^	0.88	11	11	11
NSP2	C*03:02	^ 387 ^ TILDGISQY^395^	0.73	22	22	22
NSP3	A*24:02	^726^YYTSNPTTF^734^	0.99	22	22	22
NSP3	A*24:02	^1349^NYMPYFFTL^1357^	0.98	33	22	11
NSP3	A*24:02	^816^YYHTTDPSF^824^	0.96	11	11	0
NSP3	B*44:03	^120^EEFEPSTQY^128^	0.99	11	11	0
NSP3	B*44:03	^546^QEILGTVSW^554^	0.99	22	22	0
NSP3	B*40:01	^1799^AELAKNVSL^1807^	0.98	0	0	0
NSP3	C*04:01	^734^TFDNLKTLL^742^	0.96	11	11	11
NSP3	C*07:02	^718^YYTSNPTTF^726^	0.89	22	22	22
NSP3	C*03:02	^768^MSMTYGQQF^776^	0.87	22	22	33
NSP3	C*03:02	^336^FGADPIHSL^344^	0.87	22	22	22
NSP3	C*03:02	^1436^MSNLGMPSY^1444^	0.80	11	11	11
NSP4	A*24:02	^351^FYLTNDVSF^359^	0.92	22	22	22
NSP4	B*35:01	^174^NVLEGSVAY^182^	0.97	11	11	44
NSP4	C*03:02	^ 25 ^ YLITPVHVM^33^	0.87	11	11	11
NSP4	C*03:02	^ 174 ^ NVLEGSVAY^182^	0.73	22	22	22
NSP6	A*02:01	^141^TLMNVLTLV^149^	0.92	0	0	0
NSP6	A*24:02	^84^VYMPASWVM^92^	0.91	0	0	11
NSP6	A*24:02	^115^MYASAVVLL^123^	0.90	22	22	22
NSP6	B*35:01	^156^NALDQAISM^164^	0.88	22	22	11
NSP6	B*35:01	^167^LIISVTSNY^175^	0.56	11	11	22
NSP6	C*04:01	^ 131 ^ VYDDGARRV^139^	0.89	11	11	11
NSP14	A*02:01	^321^LLADKFPVL^329^	0.94	11	11	33
NSP15	A*02:06	^181^KVDGVVQQL^189^	0.92	22	22	11
**10mer Peptides with ≤25% Homology with OC43, HKU1, and NL63**						
**Protein**	**HLA Allele**	**Sequence**	**Sore**	**OC43 % Identity with SARS-CoV2**	**HKU1 % Identity with SARS-CoV2**	**NL63 % Identity with SARS-CoV2**
Spike	A*24:02	^368^LYNSASFSTF^377^	0.89	20	20	30
Spike	A*24:02	^788^IYKTPPIKDF^797^	0.88	10	20	10
Spike	B*44:02	^95^TEKSNIIRGW^104^	0.95	10	0	10
Spike	B*35:01	^229^LPIGINITRF^238^	0.82	10	20	20
Spike	C*04:01	^932^TVYDPLQPEL^941^	0.59	20	20	10
Spike	C*07:01	^77^KRFDNPVLPF^86^	0.47	10	10	10
Membrane	A*33:03	^137^ELVIGAVILR^146^	0.65	20	20	20
Membrane	A*33:03	^177^SYYKLGASQR^186^	0.55	40	20	20
Membrane	B*40:01	^136^SELVIGAVIL^145^	0.76	10	10	20
Envelope	A*33:03	^60^SRVKNLNSSR^69^	0.48	10	30	0
Envelope	A*33:03	^29^VTLAILTALR^38^	0.41	30	20	0
Nucleocapsid	A*33:03	^140^NTPKDHIGTR^149^	0.60	20	10	20
Nucleocapsid	A*02:01	^398^ADLDDFSKQL^407^	0.44	10	30	0
Nucleocapsid	B*44:03	^321^GMEVTPSGTW^330^	0.87	0	0	0
Nucleocapsid	B*35:01	^324^VTPSGTWLTY^333^	0.68	20	20	20
Nucleocapsid	B*40:01	^322^MEVTPSGTWL^331^	0.67	10	10	0
NSP1	A*33:03	^162^NTKHSSGVTR^171^	0.74	0	0	0
NSP1	A*33:03	^68^YVFIKRSDAR^77^	0.49	30	20	10
NSP1	A*02:06	^14^VQLSLPVLQV^23^	0.45	20	10	10
NSP1	A*33:03	^15^QLSLPVLQVR^24^	0.40	10	10	10
NSP1	B*35:01	^61^LPQLEQPYVF^70^	0.81	20	20	30
NSP1	B*40:02	^112^GEIPVAYRKV^121^	0.54	30	20	20
NSP1	B*40:02	^9^NEKTHVQLSL^18^	0.49	20	20	20
NSP1	B*35:03	^61^LPQLEQPYVF^70^	0.43	20	20	30
NSP1	B*35:03	^18^LPVLQVRDVL^27^	0.41	20	20	10
NSP1	C*15:02	^7^RTAPHGHVMV^16^	0.52	30	10	10
NSP2	A*02:01	^389^ILDGISQYSL^398^	0.65	0	0	10
NSP2	A*33:03	^529^FVTHSKGLYR^538^	0.61	0	0	30
NSP2	A*02:01	^288^KLNEEIAIIL^297^	0.59	10	10	30
NSP2	A*24:02	^1^AYTRYVDNNF^10^	0.59	30	20	20
NSP2	B*44:03	^489^KEIKESVQTF^498^	0.95	0	0	10
NSP2	B*44:03	^452^EEKFKEGVEF^461^	0.89	10	10	30
NSP2	B*40:01	^344^GEQKSILSPL^353^	0.85	10	10	20
NSP2	C*03:02	^224^IAFGGCVFSY^233^	0.59	10	10	20
NSP3	A*24:02	^16^QGYKSVNITF^25^	0.79	10	20	20
NSP3	A*24:02	^1040^EYKGPITDVF^1049^	0.75	20	20	0
NSP3	B*44:03	^120^EEEFEPSTQY^129^	0.98	20	10	0
NSP3	B*35:01	^502^VPTDNYITTY^511^	0.94	0	0	0
NSP4	A*02:06	^101^FVVPGLPGTI^110^	0.65	20	10	50
NSP4	B*40:01	^97^REVGFVVPGL^106^	0.73	20	10	50
NSP5	A*24:02	^125^VYQCAMRPNF^134^	0.50	20	20	40
NSP6	A*33:03	^84^VYMPASWVMR^93^	0.53	0	0	10
NSP6	B*35:01	^43^LPFAMGIIAM^52^	0.71	0	0	10
NSP7	B*44:03	^73^EEMLDNRATL^82^	0.58	10	10	40
NSP8	A*02:01	^151^SALWEIQQVV^160^	0.68	20	30	20
NSP8	C*03:02	^13^AAFATAQEAY^22^	0.45	10	10	40
NSP14	B*35:01	^42^IPGIPKDMTY^51^	0.92	20	20	30
NSP15	A*33:03	^215^ELAMDEFIER^224^	0.57	40	20	20
NSP15	B*44:03	^169^GEAVKTQFNY^178^	0.74	10	10	10

## Data Availability

All data is available in the manuscript and its supporting files.

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
