# Peer review of "Identification of Novel Candidate CD8+ T Cell Epitopes of the SARS-CoV2 with Homology to Other Seasonal Coronaviruses"

_viruses, 2021, doi:10.3390/v13060972_

Round 1

Reviewer 1 Report

In this report the authors describe an in silico analysis of putative T cell epitopes that may be cross reactive between SARS CoV-2 and other known coronaviruses.  This is an important topic given that T cell immunity is likely to play a key role in controlling SARS CoV-2 infections and re-call against pre-exposure to related antigens. T cell responses in the context of vaccine studies and re-infection scenarios will also feature in developing effective strategies to combat the virus.

The study as presented would appear to be comprehensive and thorough in depth and scope as a predictive epitope analysis exercise. Functional data would add hugely to this topic, though studies of this kind reported here are an essential pre-requisite to these. 

Minor points:

The word 'such' is used twice in the opening two sentences of the abstract which is perhaps not the correct use of the word in either context. 

In the final parts of the discussion when referring to variants of concern, it is preferable to have consistent nomenclature and the (S African) variant 501v2 is more correctly referred to as B.1. 351. This should be altered for consistence with B.1.1.7 nomenclature.

Author Response

Reviewer #1:

Comment 1: In this report the authors describe an in silico analysis of putative T cell epitopes that may be cross reactive between SARS CoV-2 and other known coronaviruses.  This is an important topic given that T cell immunity is likely to play a key role in controlling SARS CoV-2 infections and re-call against pre-exposure to related antigens. T cell responses in the context of vaccine studies and re-infection scenarios will also feature in developing effective strategies to combat the virus.

The study as presented would appear to be comprehensive and thorough in depth and scope as a predictive epitope analysis exercise. Functional data would add hugely to this topic, though studies of this kind reported here are an essential pre-requisite to these
Minor points: The word 'such' is used twice in the opening two sentences of the abstract which is perhaps not the correct use of the word in either context.

Response: We thank the reviewer for these comments. Functional data would be extremely important to find out if the immunogenicity of these candidate CD8+ T cell responses following natural infection and following vaccination. However, such an extensive study is beyond the scope of this study. We appreciate the comments by the reviewer and will plan for these studies in future.

We have corrected the wording in the opening sentences of the abstract.

Comment 2: In the final parts of the discussion when referring to variants of concern, it is preferable to have consistent nomenclature and the (S African) variant 501v2 is more correctly referred to as B.1. 351. This should be altered for consistence with B.1.1.7 nomenclature.

Response: We thank the reviewer for pointing this out and revised these sentences accordingly.

Reviewer 2 Report

Identification of novel candidate CD8+ T cell epitopes of the SARS-CoV2 
with homology to other seasonal coronaviruses is of importance today.  The information from the comparisons will help us identify good candidates for preventing and controlling coronavirus infection. However, the study is too simple, only through the comparisons of protein sequences  SARS-CoV2, OC43, HKU1, and NL63 along with NetMHC. The reviewer has the following concerns:

1, They only analyzed HLA-A and HLA-B. Why without the work from HLA-C?

2. They obtained the conservation of proteins. Why not predict structure variation?

3. If they validate some of the results from patients in Sri Lanka, it will be better.

4. The English can be improved. There are a lot of grammar errors.

Author Response

Identification of novel candidate CD8+ T cell epitopes of the SARS-CoV2

with homology to other seasonal coronaviruses is of importance today.  The information from the comparisons will help us identify good candidates for preventing and controlling coronavirus infection. However, the study is too simple, only through the comparisons of protein sequences  SARS-CoV2, OC43, HKU1, and NL63 along with NetMHC. The reviewer has the following concerns:

Major comments

1 They only analyzed HLA-A and HLA-B. Why without the work from HLA-C?

Thank you for this important comment. We have included the analysis of HLA-C*07 allele which was found to have a frequency of over 15% in the the Sri Lankan population. We have included the analysis of  HLA-C alleles and we have included the additional data in the revised version of the manuscript.

  1. They obtained the conservation of proteins. Why not predict structure variation?

Response: We thank the reviewer for this very important question. In this study our main aim was to identify highly immunogenic CD8 T cell epitopes which are specific for SARS-CoV-2 virus and which were cross-reactive with other seasonal corona viruses (OC43, HKU1, and NL63) which were previously found in Sri Lanka. We wished to identify immunodominant candidate CD8+ T cell epitopes and then carry out functional analysis of T cell responses to find out if pre-existing T cell responses to other seasonal coronaviruses were protective or pathogenic when infected with the SARS-CoV-2 virus. Since the aim of this study was to identify novel candidate CD8+ T cell epitopes which can be used to assess the functionality and the magnitude of the T cell response, we did not carry out any predictions of the variation in the structure, which would be more relevant to antibody epitopes.

  1. If they validate some of the results from patients in Sri Lanka, it will be better.

Response: We thank the reviewer for this suggestion. In order to validate the responses to these candidate CD8+ T cell epitopes, we will have to HLA type a large population of Sri Lankan individuals and then to carry out functional T cell assays (ELISpot assays, intracellular cytokine secretion assays, dexamer/pentamer studies). Carrying out all the work mentioned above was beyond the scope of this study, and we hope to carry out these studies in future.

  1. The English can be improved. There are a lot of grammar errors.

We apologise for these errors which we have corrected in the revised version of the manuscript.